Characterization of SSR genomic abundance and identification of SSR markers for population genetics in Chinese jujube (Ziziphus jujuba Mill.)

Fu Peng-cheng 1
Zhang Yan-Zhao 1
Ya Hui-yuan 1
Gao Qing-bo 2 qbgao@nwipb.cas.cn
1 College of Life Science, Luoyang Normal University , Luoyang , China
2 Key Laboratory of Adaptation and Evolution of Plateau Biota, Northwest Institute of Plateau Biology, Chinese Academy of Sciences , Xining , China
Prentis Peter
Electronic publication date: 2016 Feb 22
Publication date: 2016
Volume: 4
Electronic Location ID: e1735
Received 2015 Nov 1; Accepted 2016 Feb 3
Copyright: ©2016 Fu et al.
Copyright year: 2016
Copyright holder: Fu et al.
License: This is an open access article distributed under the terms of the Creative Commons Attribution License, which permits unrestricted use, distribution, reproduction and adaptation in any medium and for any purpose provided that it is properly attributed. For attribution, the original author(s), title, publication source (PeerJ) and either DOI or URL of the article must be cited.
License URL: https://creativecommons.org/licenses/by/4.0/

Keywords: Genome, Jujube, Population genetics, Microsatellite, SSR abundance, SSR primers

Funding: National Natural Science Foundation of China 31400322 31270270 31400602 Joint Funds of the National Natural Science Foundation of China U1204307 This study was funded by National Natural Science Foundation of China (Grant No. 31400322, No. 31270270, No. 31400602) and the Joint Funds of the National Natural Science Foundation of China (Grant No. U1204307). The funders had no role in study design, data collection and analysis, decision to publish, or preparation of the manuscript.

==============================
Chinese jujube (Ziziphus jujuba Mill. [Rhamnaceae]), native to China, is a major dried fruit crop in Asia. Although many simple sequence repeat (SSR) markers are available for phylogenetic analysis of jujube cultivars, few of these are validated on the level of jujube populations. In this study, we first examined the abundance of jujube SSRs with repeated unit lengths of 1–6 base pairs, and compared their distribution with those in Arabidopsis thaliana. We identified 280,596 SSRs in the assembled genome of jujube. The density of SSRs in jujube was 872.60 loci/Mb, which was much higher than in A. thaliana (221.78 loci/Mb). (A+ T)-rich repeats were dominant in the jujube genome. We then randomly selected 100 SSRs in the jujube genome with long repeats and used them to successfully design 70 primer pairs. After screening using a series of criteria, a set of 20 fluorescently labeled primer pairs was further selected and screened for polymorphisms among three jujube populations. The average number of alleles per locus was 12.8. Among the three populations, mean observed and expected heterozygosities ranged from 0.858 to 0.967 and 0.578 to 0.844, respectively. After testing in three populations, all SSRs loci were in Hardy-Weinberg equilibrium (HWE) in at least one population. Finally, removing high null allele frequency loci and linked loci, a set of 17 unlinked loci was in HWE. These markers will facilitate the study of jujube genetic structure and help elucidate the evolutionary history of this important fruit crop.

Introduction

Chinese jujube (Ziziphus jujuba Mill.; 2n = 2x = 24) is an economically important species in the Rhamnaceae that is native to China (Chen & Schirarend, 2007). One of the world’s oldest cultivated fruit tree crops, according to archaeological evidence jujube was used at least 7,700 years ago in China (Liu & Zhao, 2009). Fruits of jujube have high nutritional value as well as medicinal properties, and are consumed fresh, dried or in processed form (Li et al., 2007).

Over a long period of natural evolution and artificial selection, Chinese jujube has acquired a wide range of variation, with more than 800 cultivars reported (Liu & Wang, 2009). These cultivars are distributed across China and are propagated either by rootstock grafting or as rooted cuttings (Wang, 2001). The origins of most of these cultivars are obscure because of the frequent exchange of plant material between different cultivation areas and the lack of cultivar historical documentation (Wang et al., 2014). During the frequent exchanges, the gene flow among cultivars and the gene flow between cultivars and their wild relatives are not reported. The evolutionary history of jujube cultivars thus remains unclear.

However, genetic studies based on molecular markers could offer new clues to the evolutionary history of cultivars (Sefc et al., 2000; Testolin et al., 2000; Pan et al., 2011). Simple sequence repeats (SSRs) are tandemly repeated nucleotides in DNA sequences. Because of their high polymorphism, codominance and stability, SSR markers are widely used as genetic markers to analyze genetic diversity, phylogenetic relationships and the biology of populations (Sunnucks, 2000; Barkley et al., 2006). The development of SSR markers for jujube is therefore useful for reconstruction of phylogenetic relationships among cultivars, assessment of genetic diversity, and for breeding and genetic analysis of this species.

Several studies have reported the development of SSR markers for jujube. For example, 66 expressed sequence tag-SSRs were developed from a Chinese jujube fruit cDNA library and amplified in six cultivars, but without evaluation of polymorphism (Liu et al., 2014a). Using selectively amplified microsatellites, Ma, Wang & Liang (2011) constructed a set of 25 primer pairs, and an additional 31 primer pairs were developed from SSR-enriched genomic libraries (Wang et al., 2014). Seventy-one polymorphic tri-nucleotide SSR markers were developed from assembly fruit transcriptome of jujube (Li et al., 2014). A set of 551 primer pairs were developed from jujube genome and tested in 6 jujube cultivars (Xiao et al., 2015). These primer pairs were polymorphic among different jujube cultivars and were used to infer genetic relationships among major Chinese jujube cultivars, but few of these are validated at the level of jujube populations.

Nine microsatellites were validated on the level of jujube populations in the study about the genetic diversity and population structure of Z. acidojujuba, which is considered as the ancestor of cultivated jujube (Zhang et al., 2015). However, study of population structure of jujube cultivars is rare. Cultivars often have lower genetic polymorphism levels than wild species, such as peanut (Moretzsohn et al., 2004), soybean (Li et al., 2010), apple (Zhang et al., 2012), due to diversity loss during artificial selection and vegetative propagation. Markers with high polymorphism levels in jujube cultivar populations are thus still needed for genetic and evolutionary research on Chinese jujube. The recent availability of the jujube genome sequence (Liu et al., 2014b) allows rapid identification of highly polymorphic markers for population genetics and facilitates SSR primer development.

For complementing previous data sets, we develop highly polymorphic markers in jujube cultivar populations and test them in populations as well. In this study, we examined the abundance of microsatellites with repeated unit lengths of 1–6 bp in jujube, characterized their distribution in the jujube genome, and developed 70 SSR markers from the genome sequence. After screening using a series of criteria, we finally used 20 polymorphic primer pairs to genetically characterize three jujube populations. The validated microsatellites could be used in research about the genetic structure and evolutionary history of Chinese jujube.

Materials and Methods

SSR identification and primer design

The final assembly of Chinese jujube genome spans 437.65 Mb (98.6% of the estimated size) with 321.45 Mb anchored to the 12 pseudo-chromosomes (Liu et al., 2014b). The 12 pseudo-chromosomes sequences were downloaded from GenBank (accession number: CM003114– CM003125) and scanned using MSDB v2.4 software (http://msdb.biosv.com) to identify all perfect microsatellites (Weber, 1990). The identification criteria used for mono-, di-, tri-, tetra-, penta- and hexanucleotide repeats were default in MSDB, with a minimum of 12, 7, 5, 4, 4 and 4 repeats, respectively. After analyzing each chromosome independently, all 12 chromosomes were analyzed together. Genome sequences of Arabidopsis thaliana were downloaded from GenBank (accession number: CP002684– CP002688) and comparatively analyzed against the jujube genome.

After searching the complete jujube genome, primers were designed in PRIMER v5.0 (Clarke & Gorley, 2001) using loci with repeat numbers greater than 25, for possible high polymorphism, and less than 100 for fitting with the scoring size standards. Microsatellites with mononucleotide repeats were removed when primer were designed. All primers were designed according to the following parameters: (1) product size from 100 to 400 bp and (2) primer size from 18 to 22 bp with an optimum size of 20 bp. All primers were synthesized by Tsingke Biological Technology Co. (Beijing, China). After screening using a series of criteria, parts of forward primer 5′ ends were labeled with a fluorescent dye (FAM or HEX) from Tsingke Biological Technology Co.

Plant materials and genomic DNA isolation

In this study, we tested the amplification and polymorphism of all primers using three populations: two populations (17 individuals each) of jujube cultivars ‘Bianhesuan’ and ‘Huizao’ and one population (14 individuals) of Z. acidojujuba C.Y. Cheng et M.J. Liu, the ancestor of Chinese jujube (Liu & Wang, 2009). We treated all plants of a cultivar collected from its major planting regions as a population. In addition, 10 main jujube cultivars were obtained from the Xinzheng Jujube Research Institute, Henan, China, to test the amplification feasibility of SSR markers in different cultivars. One individual was collected for each cultivar. Sample information is provided in Table S1. Total genomic DNA was extracted from dried leaves with a modified CTAB method (Doyle & Doyle, 1987), quantified using a NanoDrop 2000c spectrophotometer (Thermo Scientific, Waltham, MA, USA), and checked for quality by gel electrophoresis.

Polymerase chain reaction (PCR) and fragment analysis

PCR amplifications were carried out in 15-µl reaction volumes containing 10–20 ng template DNA, 1× PCR Buffer, 1.5 mM MgCl2, 0.4 µM of each dNTP, 0.2 µM of each primer and 1 unit of Taq DNA polymerase (Takara, Dalian, China). The PCR cycling profile consisted of an initial step of 95 °C for 5 min, followed by 35 cycles of 95 °C for 45 s, 50–55 °C (annealing temperature of each primer) for 45 s and 72 °C for 30 s, with a final extension step of 72 °C for 7 min. The PCR products were subsequently detected using a 3730XL Genetic Analyzer Sequencer (Applied Biosystems, Foster City, California). Allele sizing was performed in GENEMAPPER v4.0 (Applied Biosystems) by comparing alleles with a GeneScan-500LIZ size standard (Applied Biosystems).

Analysis of genetic diversity

The presence of null alleles, scoring errors and large allele dropout was checked using MICROCHECKER version 2.2.3 (Van Oosterhout et al., 2004).The null allele frequency (r), inbreeding coefficient (Fis) and linkage disequilibrium (LD) between all pairs of polymorphic loci were calculated using program GENEPOP 4.0 (Raymond & Rousset, 1995; Rousset, 2008). The LD was tested with 10,000 permutations. The number of alleles per locus (Na), observed (Ho) and expected heterozygosity (He), deviations from Hardy-Weinberg equilibrium (HWE) were calculated using ARLEQUIN version 3.5 (Excoffier & Lischer, 2010). The HWE was tested with 1,000,000 steps in Markov chain.

Results

Characterization of ssr distribution

The frequency and density of SSRs on each of the 12 jujube chromosomes ranged from 818.29 (chromosome 5) to 920.85 (chromosome 12) loci/Mb and from 16,035.53 (chromosome 5) to 17,750.21 (chromosome 12) bp/Mb, respectively. Analysis of 321.56 Mb of sequence data from the assembled jujube genome uncovered 280,596 SSRs covering 5.37 Mb. These SSRs included 142,795 mono-, 87,362 di-, 29,900 tri-, 15,827 tetra-, 3,237 penta- and 1,475 hexanucleotide repeats, which respectively corresponded to 46.88%, 35.12%, 10.87%, 5.22%, 1.27% and 0.73% of total SSRs. There were 373 SSRs with more than 100 repeat motifs in the jujube genome, the longest of which comprised 2,446 AT repeats. Analysis of 119.15 Mb of the A. thaliana genome revealed only two SSRs with more than 100 repeat motifs (201 AAG and 117 TCT). Average SSR sizes in jujube and A. thaliana genomes were 19.14 and 17.40 bp, respectively. Jujube genome SSR frequency and density were 872.60 loci/Mb and 16,700.42 bp/Mb, respectively, with corresponding values of 221.78 loci/Mb and 3,858.19 bp/Mb in A. thaliana.

SSR frequency and density varied with motif length. As motif length increased (from mono- to hexanucleotide repeats), the frequency of repeats decreased from 444.06 to 4.59 loci/Mb and density decreased from 7,829.71 to 121.64 bp/Mb (Fig. 1). The 10 most frequent motif types in the jujube genome were two mononucleotide (A/T, G/C), three dinucleotide (AT/TA, AG/CT, GT/AC), four trinucleotide (ATT/AAT, AAG/CTT, ATC/GAT, AAC/GTT) and one tetranucleotide (ATTT/AAAT) repeats (Table 1). When calculating the frequency of each motif type, we considered all possible types in a repeat sequence, such as two potential types (AG and GA) for the AG motif repeat and three potential types (ATT, TTA and TAT) for the ATT motif repeat. As for A. thaliana, no tetranucleotide repeats were observed among the 10 most frequent motif types. The average length, frequency and density of the 10 most frequent microsatellite motif types are provided in Table 1.

Figure 1 The density (bp/Mb) of microsatellite with repeated unit lengths of 1–6 base pairs in genome of Chinese jujube (Ziziphus jujuba Mill.) and Arabidopsis thaliana.

Table 1 The ten most frequency microsatellite motif type in genome of Chinese jujube (Ziziphus jujuba Mill.) and Arabidopsis thaliana.

No.	Motif	Average length (bp)	Frequency (loci/Mb)	Density (bp/Mb)	
	Z. jujuba	A. thaliana	Z. jujuba	A. thaliana	Z. jujuba	A. thaliana	Z. jujuba	A. thaliana	
1	A/T	A/T	17.64	14.98	436.39	112.63	7699.69	1686.89	
2	AT/TA	AT/TA	20.37	22.60	201.50	30.68	4103.91	693.38	
3	ATT/AAT	AAG/CTT	19.59	18.63	59.10	21.13	1157.87	393.79	
4	AG/CT	AG/CT	25.04	20.86	50.39	16.95	1261.91	353.66	
5	ATTT/AAAT	ATG/CAT	17.54	18.51	39.07	7.34	685.46	135.76	
6	GT/AC	AAC/GTT	24.39	17.96	19.73	5.81	481.22	104.31	
7	AAG/CTT	GT/AC	20.07	16.72	17.72	4.30	355.69	71.84	
8	G/C	AAT/ATT	16.94	19.24	7.68	3.31	130.12	63.63	
9	ATC/GAT	AGG/CCT	19.19	17.19	5.16	2.93	99.02	50.36	
10	AAC/GTT	ACC/GGT	18.90	16.80	5.05	1.99	95.52	33.41	
Notes.

Each motif type contained all the possible cases in a repeat sequence; for example, AG contained AG and GA, ATT contained ATT, TTA and TAT.

Primer design and evaluation

Taking no account of mononucleotide repeats, a total of 664 SSRs with repeat numbers greater than 25 and less than 100 were detected in jujube. We randomly selected 100 SSRs from the 664 loci and successfully designed 70 primer pairs. Screened in 2% agarose electrophoresis, 42 of the 70 primer pairs produced clear, unique amplification products of the expected size. All the 42 SSR markers showed their ability to detect polymorphisms across jujube cultivars and wild species. A set of 20 SSR markers that amplified the most easily scorable fragment polymorphisms was chosen to label with a fluorescent dye and evaluate polymorphism in the three populations. Characteristics of these 20 SSR loci are listed in Table 2. All 20 primer pairs were also amplified successfully in the 10 jujube cultivars. Among the 20 loci, Na ranged from 6 to 21, with an average of 12.8 alleles per locus (Table 2). The mean value of r among the three populations ranged from 0 to 0.261. Loci Juju18 presented too many null alleles (r > 0.15) in all populations and was eliminated from further analysis. Average Ho and He across the three populations were 0.909 and 0.701, respectively. The Na had the highest mean values (8.5) in population 1 (Z. acidojujuba) among the three populations (Table 3). Population 2 (‘Bianhesuan’) has the highest Ho (0.967) but lowest He (0.578). The Ho and He were zero at locus Juju64 in population 2. Of the 20 loci, 16 (all except Juju18, Juju57, Juju63, Juju64 and Juju68) showed significant deviation from HWE in two cultivar populations (Table 3). None loci showed significant deviation from HWE in population 1. The Fis of the three populations were −0.066, −0.490 and −0.305, respectively. Three locus pairs (Juju11 & Juju23, Juju11 & Juju25 and Juju25 & Juju42) showed signficant LD (P < 0.001) across the three populations.

Table 2 Characteristics of 20 microsatellite loci and primer pairs developed from Chinese jujube (Ziziphus jujuba Mill.) genome.

Locus	Primer sequences(5′–3′)	Repeat motif	Size (bp)	Ta (°C)	Total no. of alleles	Fluorescent dye	Location	Accession in Genebank	
Juju2	F: ACATGGAGAAATGGGATC	(AG)87	220–256	53	15	FAM	add_scaffold 107	KN813357.1	
	R: GGTTGATAGGTGGTTTGC								
Juju5	F: GGCGACGATTAGAGGAAA	(AG)36	202–226	53	7	FAM	add_scaffold 247	KN813805.1	
	R: CTGCTTGTACGGCCAGTT								
Juju10	F: AAGCGGTTGTGGTATGGG	(AG)108	147–169	53	11	FAM	add_scaffold 290	KN813315.1	
	R: TTTCTGCCACCTGCTCCT								
Juju11	F: CCACTTGCGTTACTTCTC	(CT)92	192–222	53	8	FAM	chromosome 6	CM003119.1	
	R: AATCTCGCTTTGCTCTAT								
Juju13	F: TAGGCATTTGCATGGTAT	(TC)32	190–208	53	8	HEX	chromosome 3	CM003116.1	
	R: TTGTCCGCTTTCTTGAGT								
Juju17	F: AATCGGTTACATTGCTGC	(GA)25	118–252	53	12	HEX	chromosome 6	CM003119.1	
	R: TTTCGGAGGTTACCACAT								
Juju18	F: GATGTACGGGAAAGACGG	(CT)52	268–334	53	21	FAM	scaffold 959	KN812918.1	
	R: ATCACTCCTGGTTGCTTG								
Juju23	F: CCATCCGACCACTGAAAT	(AG)29	107–163	53	20	FAM	chromosome 10	CM003123.1	
	R: CGTAAAGCACCAGCAAAA								
Juju25	F: GTACGGTATGACTCCACA	(CA)48	116–192	53	17	FAM	chromosome 3	CM003116.1	
	R: CATCCAATCACTGAAAAT								
Juju30	F: AAATGACCATCGAATCCC	(GA)25	224–260	53	11	HEX	chromosome 11	CM003124.1	
	R: CTTTGTTGTTACCCCAGA								
Juju35	F: TTGGATTAGTGTACTTGG	(TC)47	109–219	53	12	HEX	chromosome 6	CM003119.1	
	R: ACATGAGGAAACCTGGAA								
Juju42	F: CTTCAGGACGGACCAAAT	(AG)25	150–230	50	21	HEX	chromosome 8	CM003121.1	
	R: GAATGCTTCAATAAACTC								
Juju43	F: CCAAATTGCCAGGTCTAG	(AG)27	164–192	53	11	HEX	chromosome 2	CM003115.1	
	R: AACTGATCCTCCTTCGTC								
Juju52	F: TTGAAAAGGAAGGAAGAG	(TG)28	203–229	50	10	HEX	chromosome 6	CM003119.1	
	R: TGAGGATTATGAAGGGTT								
Juju54	F: GAATCCTTACATCCAATA	(TACA)37	86–158	53	6	HEX	chromosome 4	CM003117.1	
	R: ACTTACCATAATCTGTGC								
Juju57	F: ATTTATTCCTTATTGCTAGTAG	(CA)27	122–210	55	18	HEX	scaffold 844	KN812877.1	
	R: CAACCTTCTTGTAGTTATTTT								
Juju63	F: ATCAGCCAGCGTCACAAA	(TA)35	155–205	55	10	FAM	chromosome 11	CM003124.1	
	R: ATCCAAATAAGCCCACCT								
Juju64	F: ATATTGGAAACTTTCTGATC	(TC)97	108–138	53	6	FAM	chromosome 11	CM003124.1	
	R: CTGTAATACTGGGATGCT								
Juju66	F: TGGATACCGTGAAGGAAC	(GA)68	178–216	53	13	HEX	chromosome 7	CM003120.1	
	R: AGCCCATTAGAAAGCAAC								
Juju68	F: AGGCTTCAACTCTTATCC	(TAA)61	128–204	53	19	FAM	chromosome 1	CM003114.1	
	R: CCAAAACCACCACAAAAT								
Notes.

Ta annealing temperatur

Table 3 Results of initial primer screening in three populations.

One population is Ziziphus acidojujuba. CY Cheng et MJ Liu and the other two are cultivars “Bianhesuan” and “Huizao.”

Locus	Z. acidojujuba (N = 14)	‘Bianhesuan’ (N = 17)	‘Huizao’ (N = 17)	
	Na	r	Ho	Hea	Fis	Na	r	Ho	Hea	Fis	Na	r	Ho	Hea	Fis	
Juju2	10	0.000	1.000	0.846	−0.182	6	0.039	1.000	0.649*	−0.568	3	0.298	0.727	0.685*	−0.074	
Juju5	6	0.000	0.857	0.757	−0.139	3	0.000	0.941	0.544*	−0.772	3	0.000	0.667	0.665*	0.059	
Juju10	6	0.000	0.800	0.728	−0.059	7	0.000	0.941	0.695*	−0.369	6	0.231	0.455	0.749*	0.444	
Juju11	6	0.000	0.875	0.827	−0.077	3	0.000	1.000	0.569*	−0.801	4	0.051	1.000	0.720*	−0.345	
Juju13	6	0.076	0.714	0.765	0.068	5	0.000	1.000	0.386	0.000	5	0.139	0.538	0.706*	0.276	
Juju17	6	0.000	0.769	0.783	0.036	4	0.000	1.000	0.298	0.000	6	0.000	1.000	0.619*	−0.649	
Juju18	10	0.137	1.000	0.902	−0.113	4	0.400	0.667	0.363	0.273	7	0.246	1.000	0.820	−0.228	
Juju23	12	0.000	1.000	0.913	−0.100	3	0.000	1.000	0.544*	−0.889	4	0.000	1.000	0.697*	−0.455	
Juju25	11	0.000	0.786	0.884	0.115	4	0.000	1.000	0.644*	−0.581	11	0.076	0.765	0.868*	0.122	
Juju30	10	0.000	1.000	0.889	−0.116	4	0.000	1.000	0.622*	−0.639	3	0.000	0.909	0.500	−0.681	
Juju35	7	0.159	0.929	0.857	−0.087	3	0.000	1.000	0.544*	−0.889	3	0.161	1.000	0.544*	−0.889	
Juju42	12	0.037	0.857	0.915	0.066	7	0.000	1.000	0.647*	−0.572	11	0.060	0.824	0.854*	0.037	
Juju43	8	0.000	1.000	0.863	−0.154	5	0.000	0.882	0.642*	−0.391	5	0.000	1.000	0.748	−0.360	
Juju52	10	0.000	1.000	0.913	−0.100	6	0.000	0.941	0.759*	−0.249	3	0.035	0.647	0.611*	−0.060	
Juju54	5	0.000	0.857	0.754	−0.143	4	0.000	1.000	0.571*	−0.795	2	0.000	0.938	0.511*	−0.875	
Juju57	11	0.000	1.000	0.909	−0.125	5	0.000	1.000	0.609	−0.395	9	0.055	0.833	0.778	−0.084	
Juju63	7	0.100	0.900	0.844	−0.066	5	0.000	1.000	0.646	−0.440	7	0.000	0.929	0.802	−0.134	
Juju64	5	0.000	0.917	0.726	−0.228	1	0.000	0.000	0.000	–	2	0.000	1.000	0.453	−1.000	
Juju66	11	0.000	0.833	0.905	0.091	5	0.000	1.000	0.595*	−0.716	2	0.000	1.000	0.515*	−1.000	
Juju68	11	0.060	0.917	0.895	−0.017	5	0.000	1.000	0.648	−0.507	6	0.048	0.923	0.779	−0.205	
Mean	8.5	0.028	0.901	0.844	−0.066	4.6	0.022	0.967	0.578	−0.490	5.1	0.070	0.858	0.681	−0.305	
Notes.

Na total number of alleles per locus

Ho observed expected heterozygosities

He expected heterozygosities

N sample size for each population

r null allele frequency

Fis inbreeding coefficient

a Significant departure from HWE at ∗P < 0.01, respectively.

Discussion

Characterization of SSR abundance

The characterization of SSRs in our study was different from that reported by Xiao et al. (2015). This difference was caused by two reasons. One was that their study was based on scaffold sequences and analysis 396.18 Mb of sequence data (Xiao et al., 2015) and our study was based on assembled 12 chromosomes and analysis 321.56 Mb of sequence data. The other was the different identification criteria of defining SSR. Their identification criteria used for mono-, di-, tri-, tetra-, penta- and hexanucleotide repeats were a minimum of 10, 5, 5, 5, 5 and 5, respectively (Xiao et al., 2015); however, ours were a minimum of 12, 7, 5, 4, 4 and 4. We used stricter criteria for mono- and dinucleotide repeats and therefore got less SSR loci. Although the results of two studies were different, they got consistent conclusions.

We calculated the frequencies of perfect SSRs composed of 1–6-bp long motifs in jujube and A. thaliana. The abundance of certain repeat types differed between the two plants. Jujube had a much higher SSR density than A. thaliana, as well as mulberry, peach, apple, pear, grape, strawberry and Prunus mume (Xiao et al., 2015). SSRs containing A and T (A/T, AT/TA and ATT/AAT) were dominant in the jujube genome, consistent with its high AT content (66.59%, Liu et al., 2014b). High proportions of (A+ T)-rich repeats (Xiao et al., 2015) are also present in A. thaliana and other plants (Lagercrantz, Ellegren & Andersson, 1993). In protein-coding regions, the high proportion of (A+ T)-rich repeats is due to the poly(A/T) tails of densely scattered retroposed sequences and processed pseudogenes (Tóth, Gáspári & Jurka, 2000).

Microsatellites are associated with nonrepetitive DNA and have their highest frequencies in transcribed regions (Morgante, Hanafey & Powell, 2002). Besides their utility as genetic markers, SSRs have important developmental, gene regulatory and evolutionary functions (Lawson & Zhang, 2006). The function of the high-density SSRs in the complex jujube genome (Liu et al., 2014b) thus warrants further study.

The frequency of repeats in jujube decreased with increasing motif length, similar to sorghum, rice, Medicago, Populus, Brachypodium and Brassica oleracea (Iniguez-Luy, Voort & Osborn, 2008; Sonah et al., 2011). Mononucleotide repeats were the most abundant repeats. One of the 10 most abundant motif types in jujube were tetranucleotide motifs (ATTT/AAAT) which was not found in A. thaliana. This difference may be ascribed to the complexity of the jujube genome and its high SSR density, as well as its high AT content (66.59%, Liu et al., 2014b).

Primer evaluation

More alleles were detected in population 1 than in populations 2 and 3. This difference may be partly explained by the fact that population 1 consisted of a wild species whereas populations 2 and 3 were cultivars. It is relevant to the lower Fis in population 1 than in populations 2 and 3. Wild species often have higher genetic polymorphism levels than cultivars, such as peanut (Moretzsohn et al., 2004), soybean (Li et al., 2010), apple (Zhang et al., 2012). Using nine microsatellites, the average Ho and He of 31 Z. acidojujuba populations was 0.679 and 0.659, respectively (Zhang et al., 2015). Comparing the Z. acidojujuba population in our study (from Luoyang) with their population from Luoyang (EYYC), our markers identified higher Na (8.5 vs. 3.9) and higher Ho and He (0.901 and 0.844 vs. 0.683 and 0.625, respectively) (Zhang et al., 2015). The higher values hinted that our marks could uncover subtler genetic structure. Locus Juju64 uncovered only one allele in population 2, causing Ho and He to be zero, and also uncovered least alleles in other two populations, thus revealing the low polymorphism of the locus.

Out of the 70 screened primer pairs, 42 (60.0%) produced clear, unique amplification products and 20 (28.6%) displayed high levels of polymorphism. Loci Juju18 presented too many null alleles thus was eliminated. After testing in three populations, all the remained 19 polymorphic SSRs were in HWE in at least one population. For our populations, especially population 2 and 3, many loci were not in HWE because cultivars are not in a random mating system. Taking LD into consideration, the removal of one locus (i.e., Juju11 and Juju25) from each linked pair would give 17 unlinked loci in HWE.

In conclusion, we developed 17 validated microsatellite primer pairs with applicability to jujube population genetics. These polymorphic markers will facilitate the study of jujube genetic structure and gene flow and aid investigations of the evolutionary history of this important fruit crop.

Supplemental Information

Table S1 Information of plant samples used in this study

Click here for additional data file.

We thank Shan-shan Sun of the Wuhan Botanical Garden, Chinese Academy of Sciences, for providing laboratory support.

Additional Information and Declarations

Competing Interests

Author Contributions

Data Availability

The authors declare there are no competing interests.

Peng-cheng Fu conceived and designed the experiments, performed the experiments, analyzed the data, wrote the paper, prepared figures and/or tables.

Yan-Zhao Zhang performed the experiments, analyzed the data.

Hui-yuan Ya reviewed drafts of the paper, revise the manuscript.

Qing-bo Gao conceived and designed the experiments, reviewed drafts of the paper.

The following information was supplied regarding data availability:

www.figshare.com/s/fd62f4687c5c11e595ff06ec4b8d1f61

www.figshare.com/s/75aecd0a7c5f11e5a3dd06ec4b8d1f61.

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
