# Peer review of "Characterization of SSR genomic abundance and identification of SSR markers for population genetics in Chinese jujube (Ziziphus jujuba Mill.)"

_PeerJ, doi:10.7717/peerj.1735_

## Round 0.1 · original submission · Minor Revisions

Overall this new version of your manuscript is greatly improved. I still think in the introduction however you need to refer more to previous studies in the area and why your microsatellites are better or complementary to these data sets. Please also make changes in line with the suggestions of the reviewer they have provided a detailed review of your paper that will help you to improve it quality.

·

Basic reporting

This is a paper study reporting the development of SSR markers. The information given is sufficient to be able to use the markers.

Table 1. It is correct to aggregate all variants of the same repeat (for instance, ACC, CAC, CCA, GGT, GTG, and TGG), but this seems not to have been done for (A) and (T) mononucleotide repeats? The numbers are so similar, is this a mistake and should one of them be removed? In the text you refer to ‘two mononucleotide (A, T)’, this should be modified as well.

Experimental design

If jujube cultivars are multiplied vegetatively, as cuttings, what is a ‘cultivar population’?! Even the explanation in the materials and Methods does not really help. Population 1 is from the wild, and hence the numbers obtained for that population make sense. But why would you sample 10 plants of each of two cultivars for the other two populations? Do they partly consist of offspring of that cultivar?! Different genotypes but known under the same name? Mutants would not be relevant as they generally have the same SSR allele pattern. In any case, it is not surprising that the authors find fewer alleles in these plants. In addition, the assumptions behind HW equilibrium are violated here, as this clearly is not the result of random mating, so it is not surprising that some were not in HWE and that there sometimes is a large difference between Ho and He. So the interpretation of population 2 and 3 should be adjusted. This does not affect the choice of these 17 markers, as it can be based on population 1.

By the way, where do I find the results on the 12 cultivars?

Line 92. You selected SSRs with at least 25 perfect repeats, which is very long, in the hope that many alleles may exist. I would have selected trinucleotide repeats or higher, as they show fewer stutter bands and can be genotyped more reliably, as the alleles differ by more nucleotides. They generally have fewer repeats than some of the dinucleotide repeats, so by only looking at the number of repeats you would not select them. Hence my question: did you have a lot of problems with stutter bands?

Validity of the findings

Line 192-195 ‘As most microsatellites reside in regions pre-dating the recent genome expansion in many plants (Morgante, Hanafey & Powell, 2002), we propose that the jujube genome may have experienced recent expansions.’ I don’t get this. If you have more A/T containing repeats, both relatively and in absolute numbers, how can this be caused by a recent genome expansion? Besides, the other species have also experienced genome doubling/expansion events.

Line 201. The frequency of SSRs with longer repeat units always decreases; I have never seen a different pattern. Line 205. Some differences in frequency across motifs exist among plants. I would think that the relatively high frequency of AAAT and ATTT could be related to the high A/T content of the genome, but the authors do not mention that possibility here. Line 206. I have never seen a plant species with more tetranucleotide repeats that trinucleotide repeats, and I very much doubt that they exist. So the results support Lagercrantz et al. with regard to the difference between plants and animals, but this has been found in so many plant species studied since 1993 that I would skip the observation altogether.

Additional comments

Textual (the English is not always grammatically correct or sentences are incomplete):
‘high repeat numbers’ is unclear, you mean to say ‘long repeats’ to avoid confusion with the length of the repeat unit.
Line 81 ‘of the estimated’ what?
Line 176 ‘This difference was caused BY two reasons’

---

## Round 0.2 · Minor Revisions

This manuscript still needs some additional changes to be made. Please make the changes suggested by the reviewer as well as those by me below. Overall the manuscript has been improved but still needs work before it can accepted.

There are still a few issues with grammar see below.
line 76 change "on" to "at"
line 84 and throughout change "researches" to "research"
line 88 change "high" to "highly"
line 215 add "and" before "therefore"

·

Basic reporting

The present paper Characterization of SSR genomic abundance and identification of SSR markers for population genetics in Chinese jujube (Ziziphus jujuba Mill.) is a primer note presenting 17 nSSR markers for Ziziphus jujuba and its closest wild relative.

Experimental design

The methods and number of primers isolated are in line with current studies characterizing nSSR markers in plant populations, and results are presented in a very clear way. In addition, it includes a comparison with Arabidopsis in terms of genome coverage. My main concern is that I cannot see the extra value of these markers compared with microsatellite markers recently reported in other studies such as Xiao et al. 2015, who developed more than 500 primer pairs polymorphic between varieties of Z. jujuba and Zhang et al. 2015 in Z. acidojujuba. The microsatellite length and repeat number are slightly different between the two studies (L 180), but in the current status of the manuscript it is not clear what the extra value of your study is. From the introduction (L 71) and Material and Methods (L 92) it would seem that you are aiming at particularly highly polymorphic SSR, but this expectation does not seem to be fully demonstrated in the results. If this is the case, a thorough analysis comparing this and all previous studies should be provided.

Validity of the findings

L 88, Please explain how the Arabidopsis genome was compared to the Jujube genome.

L 100-102, More information about the study system should be provided. For instance, the fact that Z. acidojujuba is the wild relative of the jujube should be explained earlier in the manuscript. How many cultivars of jujube do exist (i.e. how representative of the whole jujube variation is the sampling of 10 cultivars?). How did you obtain the samples of the three populations studied? If the differentiation value of this paper is that it finds genetic variation within cultivars it would be worth explaining the total distribution of these cultivars.

L103, Please state also in the text how many samples of the 10 jujube cultivars were included.

L 193, please document the possibility that the high number of SSR found in the jujube genome might be a result of recent genome expansion. May the different genome sizes play a role here?

L183, Which conclusions of the current study are consistent with Xiao et al (2015)?

L212, It would be relevant to discuss the high inbreeding found in the two jujube populations (and not only the lower diversity) in comparison with the wild ancestor.

L217, replace “Compare” by “Comparing”.

L218, replace “marks” by “markers”.

L224-227, move to Materials and Methods.

Table 3, It would be easier to name of the species and cultivars directly on the Table rather than Populations 1, 2, and 3.

---

## Round 0.3 · accepted · Accept

I agree with your rebuttal to the concerns raised by the reviewer and agree that the value in this manuscript lies in the characterisation of these markers. The manuscript has been improved and reads well.

At production, please correct the following:
on line 121 you have inserted an extra space.
on line 240 please replace "It is relevant" with "This may be related to"